# Design of facile technology for the efficient removal of hydroxypropyl guar gum from fracturing fluid

Shiliang Xu[1,2☉‡], Mengke Cui[1☉‡], Renjie Chen[1], Qiaoqing Qiu[1], Jiacai Xie[3], Yuxin Fan[4], Xiaohu Dai[1], Bin Dong[1] *

1 State Key Laboratory of Pollution Control and Resource Reuse, College of Environmental Science and Engineering, Tongji University, Shanghai, PR China, 2 Shanghai Investigation, Design & Research Institute Co. Ltd, Shanghai, PR China, 3 CNPC Research Institute of Safety & Environmental Technology Co. Ltd, State Key Laboratory of Petroleum Pollution Control, Beijing, PR China, 4 PetroChina Xinjiang Oilfield Company, Karamay, PR China

☉ These authors contributed equally to this work.
‡ These authors share first authorship on this work.
* dongbin@tongji.edu.cn

**Data Availability Statement:** All relevant data are within the manuscript and its Supporting Information files.

**Funding:** This study was funded by the Study on Fracturing Fluid Destabilization Technology (XJYT-

## Abstract

With the increasing demand for energy, fracturing technology is widely used in oilfield operations over the last decades. Typically, fracturing fluids contain various additives such as cross linkers, thickeners and proppants, and so forth, which makes it possess the properties of considerably complicated components and difficult processing procedure. There are still some difficult points needing to be explored and resolved in the hydroxypropyl guar gum (HPG) removal process, e.g., high viscosity and removal of macromolecular organic compounds. Our works provided a facile and economical HPG removal technology for fracturing fluids by designing a series of processes including gel-breaking, coagulation and precipitation according to the diffusion double layer theory. After this treatment process, the fracturing fluid can meet the requirements of reinjection, and the whole process was environment friendly without secondary pollution characteristics. In this work, the fracturing fluid were characterized by scanning electron microscopy (SEM), Energy dispersive X-ray (EDX), X-ray diffraction (XRD) and Fourier transformed infrared (FTIR) spectroscopy technologies, etc. Further, the micro-stabilization and destabilization mechanisms of HPG in fracturing fluid were carefully investigated. This study maybe opens up new perspective for HPG removal technologies, exhibiting a low cost and strong applicability in both fundamental research and practical applications.

## 1. Introduction

With the increasing demands of modern industries, energy issues has always been a great concern to humankind [1–4]. What is worthy of more attention is that crude oil as an essential non-renewable resource occupies a significant meaning in the world's energy supplies [5, 6].

gcjsyjy-2018-JS-618). The funder provided support in the form of salaries for all authors, but did not have any additional role in the study design, data collection and analysis, decision to publish, or preparation of the manuscript. These commercial affiliations mentioned including Shanghai Investigation, Design & Research Institute Co. Ltd, CNPC Research Institute of Safety & Environmental Technology Co. Ltd and PetroChina Xinjiang Oilfield Company played a role for comments and support in our study. We confirmed that this commercial affiliation does not alter our adherence to all PLOS ONE policies on sharing data and materials.

**Competing interests:** The authors declare that we have no known competing financial interests or personal relationships that could have appeared to influence the work reported in this paper.

To realize the enhancement of crude oil productivity, the fracturing technology has been widely used in the process of crude oil extraction [7, 8].

From the first use of fracturing fluid to boost crude oil production in 1947 to the present, it has undergone a huge evolution [9]. Of late, fracturing technology was used in more than 90% of crude oil recovery operations, which had significant economic benefits in the exploration of the remaining oil potential and the exploitation of tight oil [10]. As is well-known, the greatest shortcoming of fracturing fluids is that they will cause serious formation damage, which is an increasing serious issue with detrimental effects on production enhancement [11]. Moreover, some hydrophilic organic additives are difficult to remove from wastewater because of the variety of additives in the fracturing fluid and their complicated structures, such as biocides, thickeners, corrosion inhibitors, viscosity modifiers and other chemicals [12–14]. Unfortunately, fracturing liquid could also potentially be harmful to the groundwater and human health, which has become one of the current oilfield water pollution sources, as well as secondary pollution [15–17]. More explicitly, the aforementioned issues are the bottlenecks that must be faced for crude oil production.

The total amount of fracturing flowback fluid generated by the operations has increased sharply with the large-scale development of oilfields and the frequent industrial fracturing construction operations [18, 19]. More seriously, the fracturing technology always requires copious amounts of water resources, which is still a potential waste for precious resource [20, 21]. Presently, oxidation and flocculation processes are mostly adopted to the treatment of fracturing fluid throughout the world. For example, Linden et al. demonstrated the potential of an activated sludge mixed liquor to degrade guar under typical flowback conditions [22]. Lester et al. proposed a tailored treatment approach of the aeration/precipitation (and/or filtration) combining with biological treatment (to remove dissolved organic matter) followed by reverse osmosis desalination for flowback recycling in future fracturing operations [23]. Cath et al. introduced a novel application of forward osmosis (FO) facilitating water reuse for fracturing operation and reducing the need for an additional water source [24]. Yet, these examples revealed that complicated operation, high cost and secondary pollution are difficult points that could be faced for their practical treatment of fracturing fluid. In addition to these, during the treatment process of fracturing fluid, the residual guar gum also faced the problem of filters blocking, which leaded to the failure of the treatment process due to the poor removal effect of guar gum. Up to now, few works related to the in-depth study of guar gum properties in high-salt complex water environment have been reported. How to deal with the fracturing fluid has been a very important and urgent issue, and experts in different fields are committed to solving above problems.

Guar gum, a kind of environmentally friendly natural occurring polymer, is used as thickeners in fracturing fluid [25–27]. Further, physical, chemical and biological properties of guar gum are determined by its chemical structure present on its backbone [28–30]. Generally, HPG is modified on the basis of guar gum by nucleophilic substitution reaction to introduce polar hydroxypropyl with hydrophilicity, which is usually added to the fracturing fluids and widely used in practical engineering [31]. These fracturing fluids were normally prepared by mixing in appropriate amounts of surfactants such as anionic, cationic and non-ionic etc. Among them, polymers, including polyacrylamide (PAM) or polyacrylate polymers, not only increased the viscosity of wastewater but also enhanced the difficulty of analyzing related characteristics in high-salt complex water environment. In this sense, research on the micro-stabilization mechanism and destabilization technology of HPG is crucial to the treatment of fracturing fluid. Considering the efforts undertaken so far, there is still an urgent need to come up with a low-cost, facile and efficient technology for its large-scale treatment of fracturing fluid.

Herein, we designed a facile and efficient technology of "gel-breaking, coagulation, precipitation" for effective treatment of fracturing fluids. This novel approach without any further modification and tedious steps, avoiding the complex process involving in the advanced oxidation process or microbiological method and so on. During the whole experiment, potassium persulfate (KPS), polyaluminium chloride (PAC) and polyacrylamide (PAM) were used as breaker, inorganic flocculant and organic flocculant, respectively. These reagents with obvious advantages of lower-cost and easily available were much more conducive to actual operation of oilfield. In addition, chemical composition and bonding structures of simulated fracturing fluid-0 and simulated fracturing fluid were analyzed and studied in more detail. Besides, we deeply explored and studied the destabilization mechanism of HPG on the basis of DLVO (Derjaguin–Landau–Verwey–Overbeek) theory in order to explain the aggregation and dispersion behavior of colloidal particles in fracturing fluid. Finally, through coagulation and precipitation process of simulated fracturing fluid, it remarkably indicated that HPG achieved the great removal effect by analyzing the data of flocculation efficiency and TOC removal efficiency. We believe that the facile and economical method has excellent potential in practical oilfield application and provided it candidate for comprehensive applicability.

## 2. Experimental section

### 2.1. Materials

The original fracturing fluid was collected from the Klamayi oilfield of Xinjiang, China. Hydroxypropyl guar gum (HPG) was purchased from Shanghai Yuanye Biological Technology Co., Ltd. PAM (cationic type) was supplied from Xinxiang Jinghua Water Purification Material Co., Ltd. Chitosan was provided by Shanghai Qiangshun Chemical Reagent Co., Ltd. Polyaluminium chloride (PAC), sodium sulfide ($Na_2S$) and sodium tetraborate decahydrate were obtained from Aladdin Chemical Co., Ltd, China. Sodium hydroxide (NaOH), potassium bicarbonate ($KHCO_3$), magnesium chloride ($MgCl_2$), potassium persulfate (KPS) and potassium chloride (KCl) were purchased from Sinopharm Chemical Reagent Co., Ltd. The other reagents were all of analytical reagent grade.

### 2.2. Preparation and gel-breaking process of simulated fracturing fluid

The base fluid was prepared by mixing different reagents (0.5 wt% KCl, 0.4 wt% HPG, 0.015 wt% $K_2S_2O_8$ and 0.005 wt% NaOH) and ionic mother liquor. 4 wt% borax was used as crosslinker and the crosslinking ratio of base fluid and crosslinker of 10: 1. Of these, the ionic mother liquor was the simulated ionic environment (Table 1). The specific preparation process and gel-breaking process were as follows. First, 0.5 wt% KCl was added to the ionic mother liquor. After that, HPG (0.4 wt%) was added slowly to the mixture under agitation. Then, 0.015 wt% $K_2S_2O_8$ (150 mg·L$^{-1}$) and 0.005 wt% NaOH were added to the above-mentioned

**Table 1. Formulation of ionic mother liquor.**

| Reagents | Concentration (mg·L$^{-1}$) |
|---|---|
| $FeCl_3 \cdot 6H_2O$ | 340.8 |
| $Na_2S$ | 60.8 |
| $Na_2SO_4$ | 198.8 |
| $Na_3PO_4$ | 736 |
| $MgCl_2$ | 74.1 |
| $CaCl_2$ | 2497.5 |
| $KHCO_3$ | 271.7 |

solution and then set aside for 15 min. Subsequently, the resulting solution and crosslinker were mixed homogeneously in a ratio of 10:1 by stirring magnetically. The process was carried out under the 90°C water bath condition for about 2 h (S1 Fig). After process was finished, the mixture was naturally cooled to room temperature. At this stage, pH of the mixture was approximately 6.5. Finally, the resulting supernatant liquid were named as simulated fracturing fluid (Table 2); that is, simulated fracturing fluid completed the gel-breaking process. Accordingly, the resulting supernatant liquid were named simulated fracturing fluid-0 if the breaker ($K_2S_2O_8$) was not added to the mixed solution.

## 2.3. Coagulation and precipitation process of simulated fracturing fluid

Firstly, the as-prepared simulated fracturing fluid completed the gel-breaking process through the specific steps. Subsequently, inorganic flocculant (PAC, 600 mg·$L^{-1}$) and organic flocculant (PAM, 50 mg·$L^{-1}$) were added into the same beaker containing simulated fracturing fluid in turn. Besides, inorganic flocculant (PAC, 800 mg·$L^{-1}$), organic flocculant (PAM, 50 mg·$L^{-1}$) and natural polymer flocculation (chitosan, 500 mg·$L^{-1}$) were added respectively to simulated fracturing fluid and mixed according to a set of requirements as the control groups. Then, the obtained mixture was rapidly stirred at 300 rpm for 1 min and subsequently mix evenly at 100 rpm for 10 min. After stirring, the above mixture was allowed to precipitate around 15 min.

## 2.4. Characterization

**2.4.1. Characterization of original fracturing fluid.** The original fracturing fluid was placed in the refrigerator at −80°C for more than 24 h until the sample was frozen hard. Then, the sample was freeze-dried in a high vacuum on the TENLIN FD-1B-80 freeze-drying machine until the moisture was completely removed. Finally, the sample was placed in a dryer to get fairly good preservation. The surface microstructures of the sample were observed by field emission scanning electron microscopy (FE-SEM, FEI Nova Nano SEM 450). The Bruker EDS QUA·NTAX equipped with an Energy dispersive X-ray (EDX) detector was used to characterize the chemical composition. The change of sample surface properties was investigated using Fourier transform infrared spectroscopy (FTIR, Nicolet 5700) with the wavenumber ranging from 1000 to 4000 $cm^{-1}$.

**2.4.2. Other characterization.** X-ray photoelectron spectroscopy (XPS, Thermo VG ESCA LAB 250) was employed to monitor the surface chemical composition of simulated fracturing fluid-0 and simulated fracturing fluid. The binding energy at 284.8 eV of C 1s peak was used to calibrate binding energy of all the spectra. The viscosity was tested by a rotational rheometer (Mars 60). The molecular weight was performed by High performance liquid chromatography (HPLC, Agilent LC 1100). The pH and alkalinity were measured with a FE-20 pH detector and FE28-standard apparatus (Mettlertoledo, Switzerland), respectively. The turbidity was obtained by using a portable turbidimeter (HACH-2100Q). Median particle size was determined using a Zetasizer Nano S90 (Malvern Instruments Ltd, UK). Total organic content (TOC) was determined by TOC analyzer (Shimadzu, TOC-4200).

**Table 2. Properties of simulated fracturing fluid.**

| Parameters | Values |
|---|---|
| pH | 6.50 ± 0.20 |
| Viscosity (mPa·s) | 2.09 ± 0.15 |
| Turbidity (NTU) | 81.00 ± 29.30 |
| Median particle size (nm) | 3500 ± 500 |

Average values of related research data are obtained through five parallel experiments to ensure the reliability of data such as flocculation efficiency, TOC content and so on.

The error range of relevant data were also presented in the form of error bars.

## 3. Results and discussion

### 3.1. Morphology and chemical composition

The surface morphologies of the original fracturing fluid were characterized by FE-SEM. SEM images of original fracturing fluid, at different magnifications are shown in Fig 1A–1D. The overall structure was relatively loose and the particles were unevenly distributed with the magnification of 500 X, as exhibited in Fig 1A. As presented in Fig 1B of 4000 X magnification, the surface of the sample particles was not porous, presenting a unique structure similar to stalactite. In addition, and some crystal particles can also be seen on the surface of original fracturing fluid. At higher magnification of 30000 X and 60000 X, it was obviously observed that the guar gum exhibited a sticky state in Fig 1C and 1D. EDX analysis displayed a presence of high amounts of Cl, O, Na and Ca within the surface of original fracturing fluid (Fig 1E), and EDX mapping confirms the homogeneous nature of the specimen by showing the distribution of each element (Figs 1F and S2).

The FT-IR spectrums of the original fracturing fluid and HPG are exhibited Fig 2A. The HPG exhibited characteristic broad absorption bands at 3423 cm$^{-1}$ for O–H stretching, and the presence of absorption band at 2925 cm$^{-1}$ was associated with C–H stretching vibrations. Additionally, the peaks at 1090 cm$^{-1}$ were assigned to C–OH bending vibrations [32, 33]. In case of original fracturing fluid, the broad band around 3400 cm$^{-1}$ was attributed to O–H stretching vibration [34, 35]. The absorption peaks at 1680 cm$^{-1}$ and 1550 cm$^{-1}$ were assigned to the stretching vibration of C = O and–COO–, respectively. We can also observe the C–OH stretching in the spectral regions between 1150 and 1000 cm$^{-1}$. It can be inferred that the structure of HPG had been destroyed in the original fracturing fluid. Further, a significant increase in hydroxyl may be due to breaking of the bond of carbon-oxygen bond at position 1 of galactose unit, $\alpha$-1, 6 glycosidic bond and $\beta$-1, 4 glycosidic bond, and some hydroxyl groups were also oxidized to carbonyl groups, depicted in Fig 2B [36]. The thickener in the original fracturing fluid was partially decomposed due to the influence of formation temperature and the presence of breaker. In other words, the HPG no longer existed in the form of cross-linked macromolecules. Besides, the complex component of original fracturing fluid might cause some interference to the analysis of the material structures.

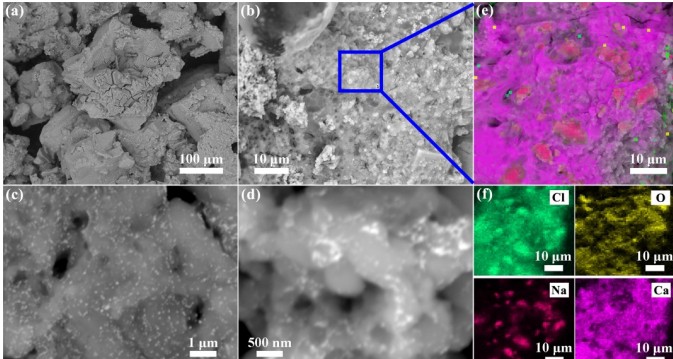

**Fig 1.** FE-SEM images of the original fracturing fluid at (a, b) low and (c, d) high magnifications. EDX Mapping results of the original fracturing fluid: (e) Mapping of all elements, (f) Mapping of Cl, O, Na and Ca.

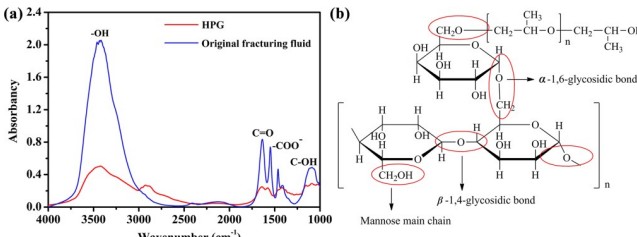

**Fig 2.** (a) FTIR spectra of the original fracturing fluid and pure HPG. (b) Schematic of broken keys for HPG.

## 3.2. Analysis of gel-breaking process

The composition of the original fracturing fluid was extremely complicated, and the structure of HPG will change when it was kept for a long period of time, affecting the properties of the original fracturing fluid. It was not conducive to the mechanism studies of HPG destabilization, etc. In this case, the as-prepared simulated fracturing fluid-0 was taken as the research object during the following mechanism study and experimental process in order to exclude some influencing factors. It is noteworthy that only when the cross-linked HPG were decomposed into small molecules can it be beneficial to the treatment of fracturing fluid and reduce formation damage in practical engineering applications. As illustrated in Fig 3, XPS measurement was used to further investigate the chemical components of the simulated fracturing fluid-0 and simulated fracturing fluid. Fig 3A and 3D display the XPS survey spectrum of the simulated fracturing fluid-0 and simulated fracturing fluid, respectively. It can be seen that the elements were detected such as C, O, N, etc. Fig 3B and 3E correspond to the bonding of C 1s of simulated fracturing fluid-0 and simulated fracturing fluid, respectively. The C 1s spectrum could be decomposed into three peaks at~284.8, 286.3 and 287.5 eV, corresponding to C–C, C–O and C = O groups, respectively [37]. After adding the breaker, the peak area ratio of C–C increased from 38% to 45%, C = O increased from 10% to 18% and C–O decreased from 52% to 37%. In addition, from the high-solution XPS spectra for N 1s (Fig 3F), whereas a new functional group of C–N at~397.8 eV appeared after the addition of breaker, as compared to that of simulated fracturing fluid-0 (Fig 3C) [38]. This indicated that the hydroxyl radicals and

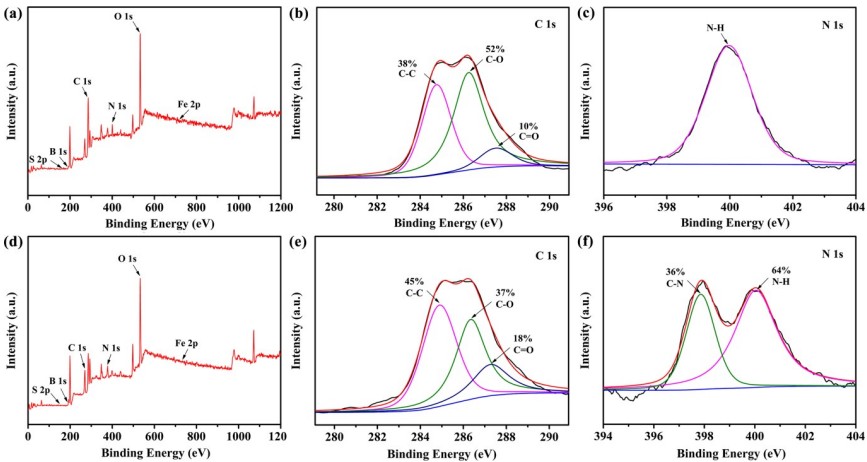

**Fig 3.** XPS full-scan spectrum of (a) simulated fracturing fluid-0 and (d) simulated fracturing fluid (after gel-breaking). High-solution C 1s XPS spectra of (b) simulated fracturing fluid-0 and (e) simulated fracturing fluid. High-solution N 1s XPS spectra of (c) simulated fracturing fluid-0 and (f) simulated fracturing fluid.

carbon-oxygen bonds were oxidized in the HPG molecules, leading to cleavage of the HPG molecular chain, which reduced the viscosity of the simulated fracturing fluid.

Guar gum is a natural high molecular polymer, which is likely to be decomposed under certain light conditions or the presence of certain microorganisms. Therefore, it is necessary to investigate the decomposition of guar gum under natural conditions. Fig 4 exhibits the changes in the MW distributions of soluble organic matter in the pure HPG, simulated fracturing fluid-0 and simulated fracturing fluid, respectively. For a pure HPG solution, the proportion of maximum MW (>1000 kDa) in the soluble organic matter significantly decreased in the first three hours, which indicated that part of the HPG should be decomposed into small molecules (Fig 4A). A strange phenomenon apparent from Fig 4A was that the proportion of maximum MW increased starting from the eighteenth hour, which inferred that samples were placed overnight resulting in molecular aggregation. As shown in Fig 4B and 4C, the low MW (1–10 kDa) of organic substance represented the largest proportion for the simulated fracturing fluid-0 and simulated fracturing fluid within 48 h. Notably, there were no maximum MW (>1000 kDa) of organic substance because of adding breaker in the simulated fracturing fluid. In general, degradation of soluble organic substance was distinct in 18–48 h in the simulated fracturing fluid-0. Furthermore, the change of MW distribution was not obvious as a function of time in the simulated fracturing fluid, and MW of approximately 70% organic substance was less than 10 kDa.

## 3.3. E-DLVO model

The wonderful process of simulated fracturing fluid from turbid to clear encouraged us to further study the destabilization performance of HPG. As is well known, the DLVO theory was developed by Derjaguin, Landau, Verwey and Overbeek [39, 40]. The theory reckons that the interaction of attractive and repulsive forces exits near the surface of a charged colloidal in an electrolyte solution [41–43]. This theory is relatively perfect in estimating trends in the mobility of numerous colloidal particles of the environment, especially in the stability of colloidal particles and the flocculating mechanism of flocculant to colloidal particles [44, 45]. Given that, we could further study about how HPG maintains a steady state in high-salt environment of fracturing fluid and seek the method of effective destabilization through the discussion of the DLVO model. This theory holds that colloidal particles are often mutually attractive and exclusive due to van der Waals interaction energy and the intersection of electrical double layer between colloidal particles, respectively [46, 47]. The van der Waals force ($V_A$) among each particle was defined by the equation [48]:

$$V_A = -A_{131}R/12H \tag{1}$$

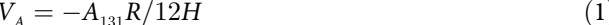

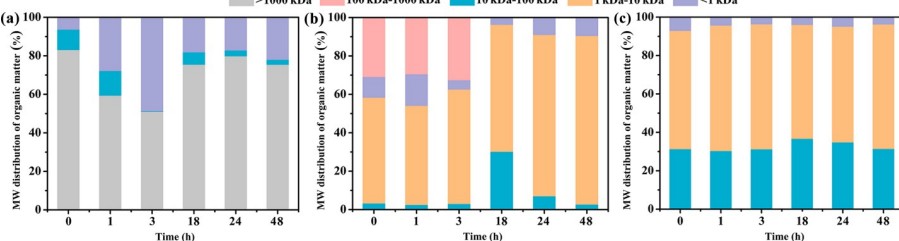

**Fig 4.** Molecular weight (MW) distributions of organic matter in the (a) pure HPG (b) simulated fracturing fluid-0 and (c) simulated fracturing fluid. Five MW fractions (<1 kDa, 1–10 kDa, 10–100 kDa, 100–1000 kDa, and >1000 kDa) were identified in the pure HPG, simulated fracturing fluid-0 and simulated fracturing fluid.

where we had the effective Hamaker constant ($A_{131}$) given by the equation below:

$$A_{131} \approx (A_{11}^{1/2} - A_{33}^{1/2})^2 \tag{2}$$

where $H$ is the operating distance of interfacial forces, $R$ is the median particle size, and $A_{131}$ represents effective Hamaker constant of particle in medium. $A_{11}$ and $A_{33}$ are the Hamaker constant of particle and medium in a vacuum, respectively. Besides, the electrostatic repulsion energy ($V_R$) was calculated according to the following equation:

$$V_R = 2\pi\varepsilon R\Phi_0^2 \ln\left[1 + \exp(-kH)\right] \tag{3}$$

where $\varepsilon$ is the permittivity of solution, $\Phi_0$ is the particle surface potential, $R$ is half of the median particle size, and $k$ is Debye constant whose reciprocal is the thickness of double layer. Moreover, Debye constant ($k$) could be defined by the equation below:

$$k = \frac{4e^2 N_A I}{\varepsilon k_B T} \tag{4}$$

where $N_A$ is the Avogadro constant, $I$ is the ionic strength, and $k_B$ is the Boltzmann constant. However, the interparticle interactions were far more complicated than ordinary colloidal dispersion systems in the actual flocculation process. Classical DLVO theory also cannot explain the aggregation and dispersion behavior of colloidal particles in these systems. Therefore, extended DLVO (E-DLVO) theory was proposed on the basis of DLVO theory. On the DLVO potential curve of particle interaction, the total energy of particles interaction ($V_T$) can be explained as follows:

$$V_T = V_A + V_R \tag{5}$$

HPG structure was similar to cellulose, and contained a large amount of -OH, -COOH and other hydrophilic groups. These hydrophilic groups absorbed several times or even dozens of times water of their own weight through hydration in an aqueous solution form a relatively stable hydration layer structure and bind water molecules within this range, which have good water retention property. Evidently, the most effective point that break its stability was damaging its hydration. The hydrophobic attraction energy between the hydrophobic colloidal particles could be attributed to the interface polar interaction particle between colloidal particles [49]. For the surface of hydrophilic particles or the surface of hydrophilic particles adsorbing chemicals, the polarizing effect of surface polar regions to adjacent water molecules formed a hydration force. The parameters of HPG E-DLVO model were summarized in Table 3.

Table 3. The parameter values of HPG E-DLVO model.

| Parameters | Values |
|---|---|
| $\Phi_0$ (mV) | $7.00 \times 10^{-3}$ |
| $\varepsilon$ (C·V$^{-1}$·m$^{-1}$) | $7.08 \times 10^{-10}$ |
| $A_{11}$ (J) | $6.30 \times 10^{-20}$ |
| $A_{33}$ (J) | $4.84 \times 10^{-20}$ |
| $R$ (m) | $6.50 \times 10^{-7}$ |
| $k_B$ (J·K$^{-1}$) | $1.38 \times 10^{-23}$ |
| $I$ (mol·L$^{-1}$) | 0.23 |
| $V_A$ | $-5.03/H$ |
| $V_R$ | $14.16 \ln(1 + e^{-0.0462H})$ |

Interaction energy curves between HPG colloidal particles are presented in Fig 5. $V_A$ and $V_R$ were all zero when the interparticle distance was far away. $V_R$ worked primarily when the interparticle distance was 20 nm. Furthermore, $V_A$ did not work until the particles overcame $V_R$, and continued to approach a certain distance, at which time $V_R$ was much larger than $V_A$. Therefore, the way to destabilize HPG colloid was that reducing the electrostatic repulsion energy between the colloidal particles, in which the colloidal particles were aggregated and agglomerated by van der Waals force, and separated from the fracturing fluid.

## 3.4. Analysis of coagulation and precipitation process

The flocculation treatment is the most common treatment technology in the area of oilfield wastewater treatment, because of its easy operation and low cost. To be specific, the flocculants realized the flocculation and sedimentation of the fracturing fluid containing massive colloidal particles and other impurities through the compression of the electric double layer, charge neutralization, adsorption bridging mechanisms and trap precipitation. More explicitly, coagulation was achieved through the flocculation of clever combination of different types of flocculants. It is widely believed that adsorption bridging and charge neutralization play a major role for organic flocculant, and these mechanisms including compression of electric double layers, adsorption bridging and trap precipitation play an important role for inorganic or natural polymer flocculant. During the flocculation process, the colloidal particles suspended in the system were destabilized, collided and condensed into larger flocs which then separated from the system by precipitation due to the addition of the flocculant. The flocculation efficiency and TOC removal efficiency of different flocculants for treating the simulated fracturing fluid are displayed Fig 6A and 6B, respectively. In contrast to other types of flocculants, PAM had high flocculation effect for the simulated fracturing fluid, which formed larger flocs and precipitated easily (Fig 6A). In order to further determine the effect of flocculation on the removal of HPG, TOC content and removal efficiency of HPG after flocculation are shown in Fig 6B. These results further confirmed that PAM possessed satisfactory flocculation effect, and the combination of PAC and PAM facilitated to achieve the best flocculation effect. From the above, it is reasonable to conclude that this flocculation method combining inorganic and organic flocculant should be selected to achieve a better flocculation effect in the flocculation process.

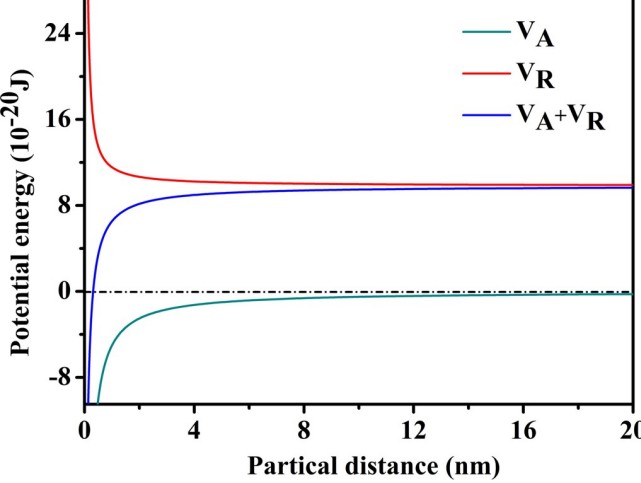

**Fig 5. Interaction energy curves between HPG colloidal particles.**

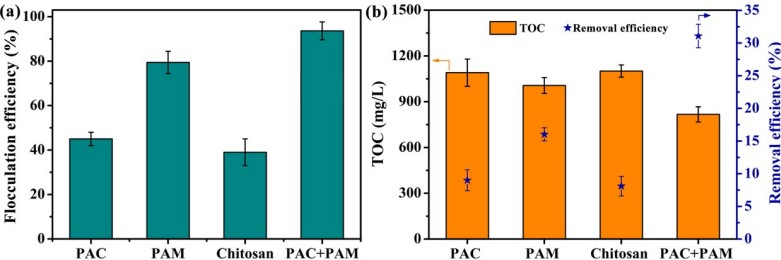

**Fig 6.** (a) The flocculation efficiency, (b) TOC content and removal efficiency of different flocculants after treating simulated fracturing fluid.

## 4. Conclusion

In summary, we designed the facile, efficient and economical HPG removal technology through a series of processes including gel-breaking, coagulation and precipitation towards the treatment of fracturing fluids. Combination of different types of flocculants achieved the HPG removal by compressing the electric double layer, charge neutralization, adsorption bridging mechanisms and trap precipitation, which make colloidal particles form floc to precipitate and separate from the fracturing fluids. The above approach is facile and low-cost easy to promote, avoiding the complicated treating process and use of complex equipment. Simultaneously, our research extends the knowledge into the explanations of aggregation and dispersion behavior between colloidal particles in fracturing fluid on the basis of DLVO theory. More importantly, this essay has argued that is the best instrument to a combination of PAC and PAM during the coagulation process. From the above, these studies thus offer a new strategy to treat the removal of HPG from fracturing fluid under the high-salt complex water environment.

## Supporting information

**S1 Fig.** The digital photos of simulated fracturing fluid (a) before and (b) after heating. (DOCX)

**S2 Fig. EDX Mapping results of the other elements of original fracturing fluid: Zr, Pu, Nb and Si.**
(DOCX)

**S1 Graphical abstract.**
(DOCX)

## Acknowledgments

We are indebted to National Engineering Research Center for Urban Pollution Control, for lab space and excellent support.

## Author Contributions

**Data curation:** Renjie Chen, Qiaoqing Qiu.

**Formal analysis:** Shiliang Xu, Bin Dong.

**Funding acquisition:** Bin Dong.

**Investigation:** Renjie Chen, Qiaoqing Qiu.

**Methodology:** Bin Dong.

**Project administration:** Jiacai Xie, Yuxin Fan.

**Resources:** Jiacai Xie, Yuxin Fan, Xiaohu Dai.

**Supervision:** Jiacai Xie, Yuxin Fan, Xiaohu Dai, Bin Dong.

**Writing – original draft:** Mengke Cui.

**Writing – review & editing:** Shiliang Xu.

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
