## [Decision Letter · Decision Letter 0]

12 Jan 2021

PONE-D-20-37253

Design of facile technology for the efficient removal of hydroxypropyl guar gum from fracturing fluid

PLOS ONE

Dear Dr. Dong,

Thank you for submitting your manuscript to PLOS ONE. After careful consideration, we feel that it has merit but does not fully meet PLOS ONE’s publication criteria as it currently stands. Therefore, we invite you to submit a revised version of the manuscript that addresses the points raised during the review process.

We look forward to receiving your revised manuscript.

Kind regards,

Yongjun Sun

Academic Editor

PLOS ONE

Journal Requirements:

2.We suggest you thoroughly copyedit your manuscript for language usage, spelling, and grammar. If you do not know anyone who can help you do this, you may wish to consider employing a professional scientific editing service.  

3. Please include your tables as part of your main manuscript and remove the individual files. Please note that supplementary tables (should remain/ be uploaded) as separate "supporting information" files.

5.Thank you for stating the following in the Acknowledgments Section of your manuscript:

"This project was funded by the Study on Fracturing Fluid Destabilization

Technology (XJYT-gcjsyjy-2018-JS-618)."

 "YES"

We note that one or more of the authors are employed by a commercial company: PetroChina Xinjiang Oilfield Company, CNPC Research Institute of Safety & Environmental Technology Co. Ltd and Shanghai Investigation, Design & Research Institute Co. Ltd

Reviewers' comments:

Reviewer's Responses to Questions

**Comments to the Author**

1. Is the manuscript technically sound, and do the data support the conclusions?

Reviewer #1: Yes

2. Has the statistical analysis been performed appropriately and rigorously? 

Reviewer #1: N/A

3. Have the authors made all data underlying the findings in their manuscript fully available?

Reviewer #1: Yes

4. Is the manuscript presented in an intelligible fashion and written in standard English?

Reviewer #1: Yes

5. Review Comments to the Author

Reviewer #1: This paper involves and put forward the hydroxypropyl guar gum (HPG) removal technologies for the efficient removal of HPG from fracturing fluid. This is a meaningful work, but significant improvements need to be made in the quality and clarity of the writing. Please check the manuscript and refine the language carefully. Also, in some places the authors need to refine and improve some of the technical content. These technical issues are listed below. If the authors make properly corrections and addressed the technical content, I think the material would be publishable.

1. Is the frozen sample (solid) still having same surface microstructures of the original fracturing fluid (fluid)? It is important for the reliability of the SEM results.

2. Experimental Section, the experiment errors and accuracy should be illustrated.

3. The HPG removal technologies in this paper should be clearly specified. It also need to show in Abstract and Conclusion.

4. The adding scheme of PAC+ PAM (polyaluminium chloride + polyacrylamide) was missing. Please check and revise.

5. What are the set of requirements for “gel-breaking, coagulation and precipitation”? On the other hand, the choice basis of potassium bicarbonate (KHCO3), PAC, PAM and natural polymer flocculation (chitosan) and the adding concentration basis of the above compounds should be elucidated.

6. HPG and crude oil can be seen as organic compound. Then the organic flocculant (PAM) has a good effect in coagulation and precipitation process, which had been proved by the results in Fig. 6. What are the modes of action of inorganic flocculant (PAC) and chitosan in coagulation and precipitation process? Please add enough discuss with good reason about their results.

6. PLOS authors have the option to publish the peer review history of their article (what does this mean?). If published, this will include your full peer review and any attached files.

Reviewer #1: No

---

## [Author Response · Author response to Decision Letter 0]

18 Jan 2021

Response to Reviewers

Dear Professor,

Thank you for giving us a chance to revise the paper and the reviewer’s comments concerning our manuscript entitled “Design of facile technology for the efficient removal of hydroxypropyl guar gum from fracturing fluid” (Manuscript number: PONE-D-20-37253). Those comments are valuable and very helpful in depth to improve the quality of the paper. We have studied comments carefully and have made corrections according to the reviewer’s suggestions.

We will be greatly honored by your approval of the corrections. Thank you very much! 

With best regards, 

Dr. Bin Dong

Response to Reviewers' comments:

The following is a point-to-point response to the reviewer’s comments.

Reviewer #1:

This paper involves and put forward the hydroxypropyl guar gum (HPG) removal technologies for the efficient removal of HPG from fracturing fluid. This is a meaningful work, but significant improvements need to be made in the quality and clarity of the writing. Please check the manuscript and refine the language carefully. Also, in some places the authors need to refine and improve some of the technical content. These technical issues are listed below. If the authors make properly corrections and addressed the technical content, I think the material would be publishable.

1. Is the frozen sample (solid) still having same surface microstructures of the original fracturing fluid (fluid)? It is important for the reliability of the SEM results.

Answer: Thank the reviewer for the good comments. We chose to use vacuum freeze-drying technology to convert the sample into a solid state because the original fracturing fluid cannot be directly used to characterize the surface microstructure. It is well-known that the vacuum freeze-drying is carried out at low temperature and vacuum environment, and the dried sample is very stable and convenient for long-term storage. Compared with other drying methods, the original structure of sample is almost unchanged since the drying process of the sample is completed in a frozen state. More importantly, the surface microstructure and organizational structure appearance of sample are well preserved, which means that the surface microstructure of freeze-dried fracturing fluid (solid) is the same as the that of original fracturing fluid (fluid).

2. Experimental Section, the experiment errors and accuracy should be illustrated.

Answer: Thank the reviewer for the good comments. The further explanations related to the experiment errors and accuracy have been added and marked red in the experimental section (Section 2.4.2).

3. The HPG removal technologies in this paper should be clearly specified. It also need to show in Abstract and Conclusion.

Answer: Thank the reviewer for the good comments. The more detailed description about HPG removal technologies have been improved and marked red in the revised abstract and conclusion.

4. The adding scheme of PAC+PAM (polyaluminium chloride + polyacrylamide) was missing. Please check and revise.

Answer: Thank the reviewer for the good comments. These mistakes have been carefully examined and corrected in the experimental section 2.3. The adding scheme of PAC+PAM have been described and marked red in detail in the revised manuscript.

5. What are the set of requirements for “gel-breaking, coagulation and precipitation”? On the other hand, the choice basis of potassium bicarbonate (KHCO3), PAC, PAM and natural polymer flocculation (chitosan) and the adding concentration basis of the above compounds should be elucidated.

Answer: Thank the reviewer for the good comments. As we all know, the outstanding fracturing operation requires that fracturing fluid not only has a high viscosity, but also can quickly complete the step of gel-breaking; at the same time, it must be economically feasible. In view of the characteristics of the above fracturing fluid and the in-depth study of gel-breaking and coagulation processes, the treatment plan of "gel breaking, coagulation, precipitation" was finally finalized. Moreover, the fracturing fluid used contains various complex inorganic salts, and potassium bicarbonate (KHCO3) is one of them in the actual oil field. More explicitly, one of the most important tasks is to highly simulate the fracturing fluid in actual environment. The selection of the types and concentrations of inorganic salt are based on the actual value in the real fracturing fluid. In addition, the most important step in our research is coagulation, and the difficulty is the selection of suitable flocculants. Therefore, we selected the most representative PAC, PAM and chitosan from the categories of inorganic, organic, and natural polymer flocculants. The above flocculants exhibited not only typical and representative properties, but also the more ideal treatment effects comparing with other similar flocculants in the preliminary experiment stage. Finally, the combination of different types of flocculants were applied to this technology to achieve the HPG removal. Meanwhile, through countless times optimization of the proportion of the flocculants added while considering cost, we got the final flocculants ratio as depicted in the experimental section. This ratio and concentration can not only meet the requirements for efficient removal of HPG, but also achieve the target of lowest cost.

6. HPG and crude oil can be seen as organic compound. Then the organic flocculant (PAM) has a good effect in coagulation and precipitation process, which had been proved by the results in Fig. 6. What are the modes of action of inorganic flocculant (PAC) and chitosan in coagulation and precipitation process? Please add enough discuss with good reason about their results.

Answer: Thank the reviewer for the good comments. The HPG removal technology is a series of designed processes including gel-breaking, coagulation and precipitation towards the treatment of fracturing fluids. It should be noted that coagulation was achieved through the flocculation of clever combination of different types of flocculants. They achieved the HPG removal by compressing the electric double layer, charge neutralization, adsorption bridging mechanisms and trap precipitation, which make colloidal particles form floc to precipitate and separate from the fracturing fluids. Reasonable combination of different types of flocculants can play a synergistic effect during the coagulation process. It is widely believed that adsorption bridging and charge neutralization play a major role for organic flocculant, and these mechanisms including compression of electric double layers, adsorption bridging and trap precipitation play an important role for inorganic or natural polymer flocculant. Some explanations and enough discussion have been added and marked red in the revised manuscript.

---

## [Decision Letter · Decision Letter 1]

17 Feb 2021

Design of facile technology for the efficient removal of hydroxypropyl guar gum from fracturing fluid

PONE-D-20-37253R1

Dear Dr. Dong,

We’re pleased to inform you that your manuscript has been judged scientifically suitable for publication and will be formally accepted for publication once it meets all outstanding technical requirements.

Kind regards,

Yongjun Sun

Academic Editor

PLOS ONE

Additional Editor Comments (optional):

Reviewers' comments:

Reviewer's Responses to Questions

**Comments to the Author**

1. If the authors have adequately addressed your comments raised in a previous round of review and you feel that this manuscript is now acceptable for publication, you may indicate that here to bypass the “Comments to the Author” section, enter your conflict of interest statement in the “Confidential to Editor” section, and submit your "Accept" recommendation.

Reviewer #1: All comments have been addressed

2. Is the manuscript technically sound, and do the data support the conclusions?

Reviewer #1: Yes

3. Has the statistical analysis been performed appropriately and rigorously? 

Reviewer #1: N/A

4. Have the authors made all data underlying the findings in their manuscript fully available?

Reviewer #1: Yes

5. Is the manuscript presented in an intelligible fashion and written in standard English?

Reviewer #1: Yes

6. Review Comments to the Author

Reviewer #1: The authors have made sufficient modifications according to the modification comments, and I suggest that this paper be accepted.

However, there are two suggestions to the authors:

1. It needs to emphasize that the application of natural polymer flocculation have a clear environment-friendly characteristic. As far as I’m concerned, it is a seductive choose over PAC or PAM.

2. For further in-depth research, it could be thought about how to tackle the environmental problems in coagulation and precipitation process.

7. PLOS authors have the option to publish the peer review history of their article (what does this mean?). If published, this will include your full peer review and any attached files.

Reviewer #1: No

---

## [Editor Report · Acceptance letter]

23 Feb 2021

PONE-D-20-37253R1 

Design of facile technology for the efficient removal of hydroxypropyl guar gum from fracturing fluid 

Dear Dr. Dong:

I'm pleased to inform you that your manuscript has been deemed suitable for publication in PLOS ONE. Congratulations! Your manuscript is now with our production department. 

Kind regards, 

on behalf of

Dr. Yongjun Sun 

Academic Editor

PLOS ONE